# Conservation of Glutathione Transferase mRNA and Protein Sequences Similar to Human and Horse Alpha Class GST A3-3 across Dog, Goat, and Opossum Species

**DOI:** 10.3390/biom13091420

**Published:** 2023-09-20

**Authors:** Shawna M. Hubert, Paul B. Samollow, Helena Lindström, Bengt Mannervik, Nancy H. Ing

**Affiliations:** 1Department of Animal Science, Texas A&M AgriLife Research, Texas A&M University, College Station, TX 77843-2471, USA; smhubert@mdanderson.org (S.M.H.); n-ing@tamu.edu (N.H.I.); 2Department of Thoracic Head & Neck Medical Oncology, University of Texas M.D. Anderson Cancer Center, Houston, TX 77030-4000, USA; 3Department of Veterinary Integrative Biosciences, School of Veterinary Medicine and Biosciences, Texas A&M University, College Station, TX 77843-2471, USA; psamollow@tamu.edu; 4Department of Biochemistry and Biophysics, Arrhenius Laboratories, Stockholm University, SE-10691 Stockholm, Sweden; helenesemail@yahoo.com; 5Faculty of Biotechnology, Texas A&M University, College Station, TX 77843-2128, USA

**Keywords:** estrogen, glutathione transferase A3-3 (GST A3-3), steroidogenesis, testes, testosterone, progesterone

## Abstract

The glutathione transferase A3-3 (GST A3-3) homodimeric enzyme is the most efficient enzyme that catalyzes isomerization of the precursors of testosterone, estradiol, and progesterone in the gonads of humans and horses. However, the presence of GST A3-3 orthologs with equally high ketosteroid isomerase activity has not been verified in other mammalian species, even though pig and cattle homologs have been cloned and studied. Identifying *GSTA3* genes is a challenge because of multiple *GSTA* gene duplications (e.g., 12 in the human genome); consequently, the *GSTA3* gene is not annotated in most genomes. To improve our understanding of *GSTA3* gene products and their functions across diverse mammalian species, we cloned homologs of the horse and human *GSTA3* mRNAs from the testes of a dog, goat, and gray short-tailed opossum, the genomes of which all currently lack *GSTA3* gene annotations. The resultant novel *GSTA3* mRNA and inferred protein sequences had a high level of conservation with human *GSTA3* mRNA and protein sequences (≥70% and ≥64% identities, respectively). Sequence conservation was also apparent for the 12 residues of the “H-site” in the 222 amino acid GSTA3 protein that is known to interact with the steroid substrates. Modeling predicted that the dog GSTA3-3 may be a more active ketosteroid isomerase than the corresponding goat or opossum enzymes. However, expression of the *GSTA3* gene was higher in liver than in other dog tissue. Our results improve understanding of the active sites of mammalian GST A3-3 enzymes, inhibitors of which might be useful for reducing steroidogenesis for medical purposes, such as fertility control or treatment of steroid-dependent diseases.

## 1. Introduction

It is widely appreciated that reproduction of vertebrate animal species centers around sex steroid production by the gonads. While driven by peptide hormones from the hypothalamic/pituitary axis, production of testosterone by the Leydig cells of the testis and estradiol and progesterone by the granulosa cells and corpora lutea of the ovary are required for successful production of offspring [1,2,3]. Testosterone, estradiol, and progesterone act in reproductive tissues by binding to specific receptors which regulate the coordinated expression of genes important to male and female reproduction [4,5].

Testosterone, estradiol, and progesterone are produced from cholesterol by a series of enzymatic modifications [6,7]. This reaction sequence differs among species: rodents utilize the ∆^4^ pathway, whereas humans and other larger mammals utilize the ∆^5^ pathway [7]. The biosynthetic pathway for testosterone in human and horse testes involves the alpha class glutathione transferase A3-3 (GST (Abbreviations used: GST glutathione transferase; ∆5-AD delta-5 androstenedione; SF1 steroidogenic factor 1; RT reverse transcription; PCR polymerase chain reaction; cDNA complementary DNA) A3-3) as a ketosteroid isomerase that acts as a homodimer to convert ∆^5^-androstene-3,17-dione (∆^5^-AD) to ∆^4^-androstene-3,17-dione, the immediate precursor of testosterone [8,9,10,11,12,13,14,15,16]. The human GST A3-3 enzyme is 230 times more efficient at this isomerization than the hydroxy-∆^5^-steroid dehydrogenase, 3β-hydroxysteroid delta-isomerase 2 (HSD3B2), the enzyme to which this activity previously had been attributed [2]. Estradiol synthesis is likewise dependent upon GSTA3-3 in humans and horses because testosterone is further converted to estradiol by aromatase [7]. GST A3-3 also participates in the synthesis of progesterone in female mammals utilizing the ∆^5^ pathway by isomerizing ∆^5^-pregnene-3,20-dione to ∆^4^-pregnene-3,20-dione (i.e., progesterone).

The *GSTA* class of genes is best characterized in humans. It is represented by 12 genes and pseudogenes with almost indistinguishable sequences. These arose from gene duplications and, in humans, are clustered in a 282 kbp region on chromosome 6 [17]. Because of the redundancy of GSTA genes, the *GSTA3* genes are inexactly identified, if identified at all, in the current annotations of the genomes of most mammals. This gap in knowledge limits studies of *GSTA3* gene expression because standard *GSTA3* mRNA and protein detection reagents crossreact with other four gene products in the GSTA family, which are ubiquitously expressed [11].

Inhibitors of the GST A3-3 homodimeric enzyme and factors that decrease *GSTA3* gene expression inhibit the synthesis of sex steroid hormones. In extracts of the steroidogenic JEG-3 cell line of placental origin, two specific inhibitors of GSTA3-3 (ethacrynic acid and tributyltin acetate) decreased isomerization of ∆^5^-AD with very low half maximal inhibitory concentrations (IC_50_) values of 2.5 µM and 0.09 µM, respectively [18]. In addition, transfection of this cell line with either of two different small, interfering RNAs decreased progesterone synthesis by 22% to 30%. In stallions treated with dexamethasone, a glucocorticoid drug used to treat inflammation, testicular levels of *GSTA3* mRNAs decreased by 50% at 12 h post-injection, concurrent with a 94% reduction in serum testosterone [19,20]. In stallion testes and cultured Leydig cells, dexamethasone also decreased concentrations of steroidogenic factor 1 (*SF1* or *NR5A1*) mRNA [21]. In a human genome-wide study of the genes regulated by SF1 protein, chromatin immunoprecipitation determined that the SF1 protein bound to the *GSTA3* promoter to up-regulate the gene [13]. Notably, the SF1 transcription factor up-regulates the expression of several gene products involved in steroidogenesis [2]. These data indicate that there are reagents and pathways by which *GSTA3* gene expression and steroidogenesis can be altered in vivo. This could be useful for reducing steroidogenesis for fertility control or other medical purposes, such as reducing tumor growth in advanced prostate cancer [22].

The equine and common marmoset GST A3-3 enzymes are structurally similar to the human enzyme, and they match or exceed the very high steroid isomerase activity of the latter [15,23]. Remarkably, the phylogenetically more closely related GSTA3 homologs from the gonads of domestic pigs (*Sus scrofa*) and cattle (*Bos taurus*) yielded enzymes that differed greatly as ketosteroid isomerases [10,24]. For reasons yet to be determined, the pig enzyme was highly active and showed 70-fold higher activity than the cow enzyme [15].

In the current work, *GSTA3* mRNA was cloned from the testes of the dog and goat (eutherian mammals), and gray short-tailed opossum (metatherian mammal). The purposes were (1) to compare mRNA and protein sequences with the human and horse counterparts to search for more highly active ketosteroid isomerases, (2) to generate GSTA3-specific reagents for future studies of the regulation of *GSTA3* genes, and (3) to understand the scope of the GSTA3-3 function in steroidogenesis across a range of mammalian species in which reproduction is important to humans. Information from these additional mammalian species will expand our knowledge of GSTA3 structure and function across a considerably broader phylogenetic range than has previously been examined (Figure 1).

## 2. Materials and Methods

### 2.1. Materials

The chemical reagents were from Sigma-Aldrich (St. Louis, MO, USA) unless otherwise noted.

### 2.2. Methods

#### 2.2.1. Testis Tissue Samples and RNA Preparation

All animal procedures were approved by the Texas A&M University Institutional Animal Care and Use Committee. Adult testes used in this study were obtained from Texas A&M University-owned research animals; specifically, from a large-breed hound (*Canis lupus familiaris*), a goat (*Capra hircus),* and a gray short-tailed opossum (*Monodelphis domestica*). Additional tissues (adrenal gland, liver, small intestine, ovary, ovary with follicles, spleen, heart, skeletal muscle, kidney, uterus, mammary gland, cerebrum, and hypothalamus) were collected from two female large-breed hounds.

Total cellular RNA was extracted from each testis with the Tripure reagent (Sigma-Aldrich; St. Louis, MO, USA) protocol. The concentration and purity of the RNA were assessed using a Nanodrop spectrophotometer.

#### 2.2.2. Reverse Transcription

Reverse transcription reactions (25 µL) for the production of cDNA were run with each RNA sample (500 ng) using random octamer primers (312 ng), dT_20_ primers (625 ng), dNTPs, DTT, first strand buffer, Superasin and Superscript II Reverse Transcriptase (RT), following the manufacturer’s instructions (ThermoFisher Scientific; Waltham, MA, USA). The RT reactions incubated in an air incubator at 42 °C for three hours. The RT enzyme was inactivated by incubating at 70 °C for 15 min.

#### 2.2.3. Cloning GSTA3 cDNAs into the Vector pCR2.1

Nested PCR was utilized to selectively amplify the *GSTA3* mRNA targets to increase the sensitivity and fidelity of amplification. Primers (Table 1) were designed from BLAST analyses with human and horse *GSTA3* mRNA sequences (GenBank accession numbers NM_000847.1 and KC512384.1, respectively) to identify related dog, goat, and opossum sequences for cloning cDNAs into the general use vector pCR2.1 (TA Cloning Kit; ThermoFisher Scientific). The primers were synthesized by Integrated DNA Technologies (Skokie, IL, USA). PCR reactions (50 µL) were performed with Ex Taq enzyme (Takara Bio Inc.; Mountain View, CA, USA) in a GeneAmp PCR System 9600 (PerkinElmer; Norwalk, CT, USA). Primary PCR reactions (50 µL) were set up with outside primers and 1 µL of the appropriate RT reaction. Cycling parameters were 30 cycles of 94 °C for 15 s, 45 °C annealing for 30 s, and 72 °C for one minute, followed by a one-time hold at 72 °C for five minutes. Secondary PCR reactions (50 µL) were set up with inside primers and 5 µL of the primary PCR reactions as a cDNA template source instead of using the RT reaction. Cycling parameters were maintained for the secondary PCR. The PCR products were run out on both a 1% agarose gel and a 0.8% low melting temperature agarose gel. Ethidium bromide staining of DNA bands was visualized under UV light. This procedure confirmed the presence of the intended PCR products of 670 to 700 base pairs in all of the secondary PCR reactions.

*GSTA3* cDNA bands were cut from the 0.8% low melting temperature agarose gels and melted at 70 °C for 10 min. Ligation of the cDNA inserts to pCR2.1 was performed according to the kit manufacturer’s instructions (TA Cloning kit; ThermoFisher). These ligation reactions were then utilized in transformation of *E. coli* competent cells. After incubation in SOC broth for 1 h at 37 °C, transformations were plated on LB-ampicillin agar containing isopropyl β-D-1-thiogalactopyranoside and 5-bromo-4-chloro-3-indolyl-β-D-galactopyranoside for blue-white colony selection. Plates were incubated overnight at 37 °C. Selected white colonies were cultured overnight in LB-ampicillin broth and plasmid DNA was purified using the Qiagen QIAprep Spin Kit (Germantown, MD, USA). *GSTA3* cDNAs were released from the pCR2.1 plasmids by digestion with EcoRI before being analyzed by gel electrophoresis. Plasmid samples that demonstrated the expected 670 to 700 base pair bands were sequenced.

#### 2.2.4. Sequencing and Structure Analyses

The *GSTA3* cDNAs inserts in the pCR2.1 vectors were amplified by PCR through the use of Big-Dye mix (Applied Biosystems, ThermoFisher Scientific) using the M13 forward and M13 reverse primers (5′-TGTAAAACGACGGCCAGT-3′ and 5′-CAGGAAACAGCTATGAC-3′, respectively). After PCR amplification, G-50 spin columns were used to remove unincorporated nucleotides. Samples were sequenced at the Texas A&M University Gene Technologies Laboratory on a PRISM 3100 Genetic Analyzer mix (Applied Biosystems, ThermoFisher Scientific). Each sequence was then compared to the NCBI GenBank database records for the human and horse *GSTA3* mRNA sequences (GenBank accession numbers NM_000847.1 and KC512384.1, respectively). The basic local alignment search tool (BLAST) on the NCBI website was also utilized to check for alignments to reference mRNA sequences of other species. For each species, the sequence with the highest identity to the NCBI *GSTA3* mRNA reference sequences was translated to its amino acid sequence through the use of the ExPASy translate tool (https://web.expasy.org/translate/ (accessed on 28 August 2023)). Sequence validation of *GSTA3* mRNAs and proteins was obtained by locating the five amino acid residues that have been identified as key to the activity of the human GST A3-3 enzyme [9,26]. Following this analysis, *GSTA3* mRNA sequences were submitted to NCBI GenBank. Accession numbers for these sequences are given in the Results and Discussion section. Sequence alignments were made with Clustal Omega [27].

Homology modeling of the 3-D structures of the novel GST A3-3 proteins was made using the program MODELLER [28] (version 10.4, University of California San Franscisco, CA, USA) which calculates a model based on the known structure of a homologous protein. The template used was the crystal structure of human GST A3-3 with glutathione and ∆^4^-androstene-3,17-dione bound in the active site (PDB code 2vcv). The model of dog GST A3-3, shown in Section 3.4, had the following scores: GA341:1.00; zDOPE: −1.80; Estimated RMSD: 0.894; Estimated Overlap (3.5 Å): 0.950. The structure was depicted in Chimera 1.17.1 [29].

#### 2.2.5. Cloning GSTA3 cDNA in the Bacterial Expression Vector pET-21a(+)

Another set of optimized primers was created for cloning into the bacterial expression vector pET-21a(+) (EMD Millipore; Billerica, MA, USA). For directional in-frame cloning into pET-21a(+), the EcoRI (GAATTC) and XhoI (CTCGAG) restriction sites were added to the 5′ ends of the inside sense and antisense primers, respectively (Table 2). Nested PCR was performed as described in Section 2.2.3. Secondary PCR samples along with the pET-21a(+) vector were digested with EcoRI and XhoI endonucleases. cDNA and plasmid vector DNA purified by electrophoresis on an 0.8% LMT agarose gel were ligated. After transformation, colony growth in broth, and plasmid mini-preparation, the plasmid cDNAs were sequenced with the T7 promoter and the T7 terminator primers (5′-TAATACGACTCACTATAG-3′ and 5′-GCTAGTTATTGCTCAGCGG-3′, respectively). Sequence analysis was as described in Section 2.2.4, and included confirming in-frame insertion of the sense strand sequence into the pET-21a(+) vector for protein expression in bacteria.

#### 2.2.6. Quantitative Reverse Transcription-PCR

Reverse transcription for the 13 female dog tissues and the male testis was performed as described in 2.2.2 except that only 100 ng of RNA was used. Real time PCR was performed in triplicate with 1 µL of 4-fold diluted cDNA, Power SYBR Green PCR Master Mix (Applied Biosystems; Foster City, CA, USA) and GSTA3-specific primers: Sense ACATCCACCTGGTTGAACTTCTCTACT; Antisense CTGGTTTTCAGGGCCTTCAG. The 96-well plates were run on an ABI 7900HT Fast Real Time PCR System for an initial denaturation of 95 °C for 15 s, then 40 cycles of 95 °C denaturation for 15 s and 60 °C annealing and extension for one minute. This was followed by a 20 min dissociation curve to assess the single PCR product’s melting temperature. The Cts were averaged for each tissue. Linear values were generated by 2^-(“tissue X” Ct- Highest Ct). Linear values presented are relative to the lowest expressor (uterus) set at 1.0.

## 3. Results

### 3.1. Alignment of the Cloned GSTA3 mRNA Sequences to Those of Human and Horse GSTA3 mRNAs

Figure 2 shows the alignments of the newly cloned dog, goat, and gray short-tailed opossum *GSTA3* mRNA sequences to the known *GSTA3* mRNA sequences of human (GenBank accession number NM_000847.4), marmoset (XM_002746649), and horse (KC512384.1). As expected, there is a high degree of conservation of the GSTA3 mRNA sequence across all species. The GenBank accession numbers of the newly cloned *GSTA3* cDNAs are KJ766127 (dog), KM578828 (goat), and KP686394 (gray short-tailed opossum). The overall percentages of identical residues at each position were compared pairwise between each *GSTA3* mRNA, and are presented in Table 3. Residue identities declined approximately as expected according to the phylogenetic relationships among the species examined. Compared to the human *GSTA3* mRNA, for example, the marmoset *GSTA3* mRNA has the highest sequence identity to that of the human, with 94% identical bases. The new dog and goat GSTA3 sequences had similar identities to each other and to human and horse GSTA3 mRNA sequences. The *GSTA3* mRNA of the gray short-tailed opossum, the only metatherian representative, showed greater divergence, with identities to the *GSTA3* mRNAs of the five eutherian species examined ranging from 68% to 71% identical bases.

### 3.2. Alignment of the Predicted GSTA3 Protein Sequences from Six Mammals

The amino acid sequences of the GSTA3 proteins were inferred from the cloned *GSTA3* mRNA sequences. They are aligned with the human and horse GSTA3 protein sequences (GenBank accession numbers NP_000838.3 and AGK36275.1, respectively) in Figure 3. One-hundred ten (110) of the 222 total amino acid positions (50%) had identical residues across the GSTA3 proteins of all six species. Similarly, 56 positions (25%) had highly similar residues and 11 positions had less similar residues (5%) within the six species’ GSTA3 proteins. The rest of the residue positions (44 or 20%) were not similar across all six GSTA3 proteins. The conservation of amino acid residues between GSTA3 proteins of different species appears to be fairly consistent across the length of the proteins. The percentage of identical amino acids in the various positions was calculated between pairs of aligned GSTA3 protein sequences of all species (Table 4). The degrees of conservation of the new dog and goat GSTA3 protein sequences to each other and to human, marmoset, and horse GSTA3 proteins were similar. The opossum GSTA3 protein exhibited weaker conservation compared to the GSTA3 proteins of the eutherian species.

### 3.3. Conservation of Amino Acids Critical for Steroid Isomerase Activity and Binding of the Cofactor Glutathione

H-site residues for steroid binding, as well as G-site residues for glutathione binding, occur throughout the length of the GSTA3 primary structure (Figure 3). The G-site residues for binding the cofactor glutathione in human GSTA3 are Tyr9, Arg15, Arg45, Gln54, Val55, Pro56, Gln67, Thr68, Asp101, Arg131, and Phe220 [15]. Compared to human proteins, the G-site residues were identical or conservatively changed among the GSTA3 proteins of the five other species we examined, with the exception of Arg45 or Lys45, replaced by Ile45 in the opossum enzyme (Figure 3). Most of the H-site residues are nearly identical across all of the species. The five critical amino acids were defined as having high ketosteroid isomerase activity in the human GSTA3-3 homodimeric enzyme, compared to the low activity of human GST A2-2 [9], and are marked with “*” in Table 5 column 1. These are identical or similar to the human GST A3-3 residues in the three new sequences, with the exception of more bulky residues in position 208. Table 5 shows the H-site residues in the GSTA enzymes expressed in pig (“GST A2-2” in [10]) and bovine (“GST A1-1” in [24]) steroidogenic tissues and comparisons against the ketosteroid isomerase activities of equine and human GSTA3-3 enzymes [15]. (The pig enzyme was the second porcine GST characterized and was therefore named GST A2-2 in spite of being the functional equivalent of human GST A3-3 [10]). Both pig GSTA2-2 and bovine GSTA1-1 enzymes are similar to the dog, goat, and opossum enzymes in having more bulky residues at position 208 than do the human and equine GSTA3-3 enzymes. This may be part of the reason that the pig and bovine enzymes show 27% and 0.38% of the equine GST A3-3 activity with the substrate ∆^5^-androstene-3,17-dione, respectively [10]. The divergent residues in position 108 of the dog, goat, and opossum GST A3-3 and the bovine GST A1-1 proteins are probably also relevant to isomerase activities based on the modeled structure.

### 3.4. Modeling the Structures of the Newly Described GST A3 Sequences

The crystal structure of human GST A3-3 in complex with Δ^4^-androstene-3,17-dione has been determined [30] and was used to model the structures of the homologous novel GST proteins. All GST A3-3 proteins appear to fold in a similar manner, and could possibly accommodate the steroid substrate in a productive binding mode without major clashes with the H-site residues. The dog GST model shows the most favorable interactions with the steroid. The high-activity human and equine GST A3-3 enzymes have smaller residues in position 208, Ala208 and Gly208, respectively (Table 5). The modeled opossum GST A3-3 enzyme indicates that its Met208 clashes with the steroid, unless the overlap can be avoided by a rotation of the Met sidechain. The goat GST A3-3 has Thr208 in this highly variable position, and this residue is smaller than Leu208 in the dog enzyme. It is noteworthy that all H-site residues of goat GST A3-3 are identical with those of the cattle enzyme (Table 5). The latter has been expressed and displays a very low steroid isomerase activity [24], possibly indicating low activity for the goat enzyme as well. On the basis of the modeling using the high-activity human GST A3-3 in complex with androstenedione [30], the dog GST A3-3 appears as the most likely one to have steroid isomerase activity of the three new enzymes (Figure 4). In comparing the H-site residues of the human GST A3-3 to the corresponding dog residues, all but four are identical between the two proteins. Ala208 in the human enzyme is replaced by the larger Leu208 in the dog enzyme, which may interfere with the D-ring of the steroid substrate. The highly active horse GST A3-3 features the smallest residue Gly208 (Table 5). The replacement of Leu108 by Val108 is presumably without consequence, since the sidechain points away from the steroid. Substitutions of Leu107 by Met107 and the C-terminal Phe222 by Ile222 are probably compatible with the binding of the steroid substrate since the sidechains are flexible. In summary, the nature of residue 208 may hamper the activity of the dog enzyme in the steroid isomerase reaction.

### 3.5. Expression of Dog GSTA3 mRNA in Dog Tissues

Quantitative reverse transcription-PCR was used to measure the expression of the GSTA3 gene in 14 tissues from adult dogs. The highest expression was found in the liver, which was six-fold greater than levels in testes (Figure 5). The GSTA3 mRNA concentrations in testis were four-fold higher than in the adrenal gland and ovaries. Small intestine and skeletal muscle had GSTA3 gene expression levels similar to gonads. This expression pattern of the dog GSTA3 is quite different than expression in horse tissues, which had the highest levels in testis and adrenal gland, which were four-fold greater than in liver [15].

## 4. Discussion

Complementary DNA were cloned from *GSTA3* mRNAs in testes from the dog, goat, and gray short-tailed opossum. Given that the sequences of the *GSTA1, GSTA2*, and *GSTA3* mRNAs are highly conserved [10], cloning of *GSTA3* mRNA was difficult, even from testes, which has a distinctively high level of *GSTA3* gene expression [15]. The PCR primers used here were designed to have their 3′ ends bind to codons or untranslated sequences that are unique to the *GSTA3* mRNA. Interestingly, there are no reagents for nucleic acid hybridization or antibody immunodetection that can distinguish *GSTA3* gene products from those of *GSTA1* and *GSTA2* genes, both of which are ubiquitously expressed. The *GSTA3* mRNA can only be specifically amplified using PCR primers, such as those designed here and previously [11,19]. For example, to localize *GSTA3* gene expression to Leydig cells within horse testes, we developed an in situ reverse transcription-PCR protocol that was specific for *GSTA3* mRNA [19].

The human GSTA3-3 homodimeric enzyme has well characterized catalytic efficiency with substrates Δ^5^—androstene-3,17-dione and Δ^5^—pregnene-3,20-dione, which is second only to the equine GSTA3 enzyme of the six GSTA proteins that have been characterized at the time of this writing [15]. The H-site residues of the GSTA3 proteins are critical for positioning the steroid substrate to undergo the ketosteroid isomerization. The amino acids required for that activity were initially determined experimentally for human GSTA3-3 [9,24]. The human and horse GSTA3 proteins have very high conservation of H-site residues (Table 5). These differ from related GSTA enzymes, such as human GSTA2-2, which have very little activity [15]. When activity data become available for the dog, goat, and opossum enzymes, we may be able to relate enzymatic activity differences to specific amino acids or combinations of them. It will also be instructive to compare the steroid isomerase activities of the newly cloned GSTA3-3 enzymes with those of the bovine [24] and pig [10] cognates of the human GSTA3-3, of which, in particular, the bovine enzyme has significantly lower steroid isomerase activities than the human and horse GSTA3-3 enzymes [15].

As a recently discovered enzyme in steroid hormone biosynthesis, the GSTA3-3 enzyme and its corresponding mRNA sequence lack extensive functional data across species. The mechanism of GSTA3-3 action, and regulation of the *GSTA3* gene, must be more fully characterized in order to better understand the roles they play in steroid hormone biosynthesis and the possible implications of impairment of the enzyme’s function. Our work contributes to this objective through identification of the *GSTA3* mRNA coding sequences in multiple species, as well as provision of *GSTA3* clones in the expression vector pET-21a(+), which can deliver GST A3-3 proteins to be evaluated for their activities as steroid isomerases. Evaluation of these enzymes will enable quantitative comparisons of the steroid isomerase activities among the species investigated, and enable further investigation of the effects of endogenous compounds, such as follicle-stimulating hormone, luteinizing hormone, testosterone, estradiol, and glucocorticoids, as well as pharmaceuticals such as phenobarbital or dexamethasone, on expression of the enzyme. For instance, expression of the *GSTA3* gene was down-regulated along with testosterone synthesis in testes of stallions treated with dexamethasone [19,20], whereas in cultured porcine Sertoli cells, follicle-stimulating hormone, testosterone, and estradiol increased *GSTA* gene expression [31]. In addition, inhibitors of the GSTA3-3 enzyme have been identified that may be medically useful for reducing fertility or progression of steroid-dependent diseases [15]. These include the GST A3-3 enzyme inhibitors ethacrynic acid or tributyltin [32], or the small interfering RNAs that reduce *GSTA3* gene expression [18].

## 5. Conclusions

The three new *GSTA3* mRNA and protein sequences we report extend our comparative knowledge of the GST A3-3 enzymes of diverse mammalian species. Moreover, the information and reagents we have generated will help to facilitate future studies to characterize the GST A3-3 enzymes more generally. In addition, they may augment studies of the regulation of the expression of *GSTA3* genes in steroidogenic and other tissues. Mechanistic data of this kind can enable development of new approaches for manipulating GSTA3-3 enzyme activities in vivo for medical purposes. These could include inducing fertility interruptions in animal species important to humans or developing novel pharmacological treatments for steroid-dependent diseases in humans and other animals. It is essential that the new treatments that are prescribed are both effective and specific. It is, therefore, crucial that researchers continue to investigate steroidogenesis and its regulation in order to generate more complete mechanistic knowledge.

## Figures and Tables

**Figure 1 biomolecules-13-01420-f001:**
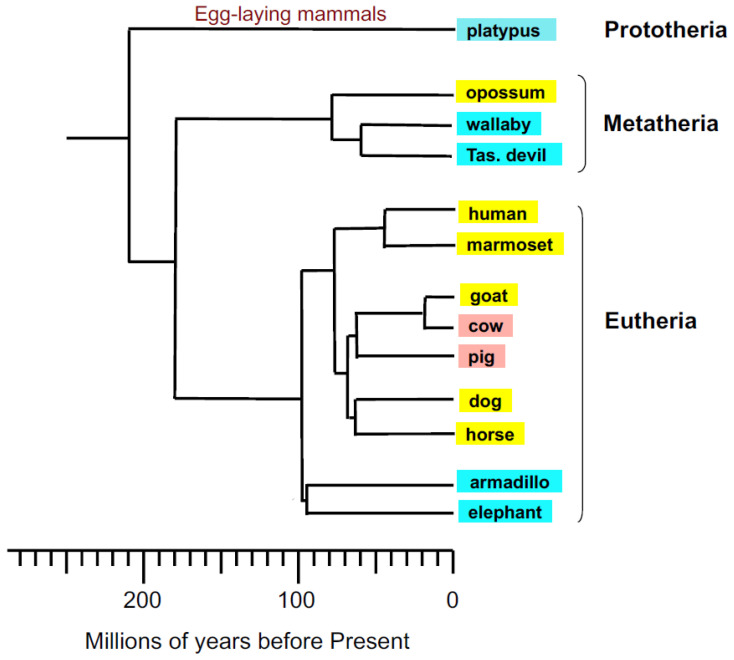
Phylogenetic relationships of among mammalian species used in this study (adapted from [25,26] with refinements from WJ Murphy, personal communication). Species investigated in this report and those used as references are highlighted in yellow; additional species with reported GSTA ketosteroid isomerase data are shown in pink; species included for phylogenetic context only are shown in blue.

**Figure 2 biomolecules-13-01420-f002:**
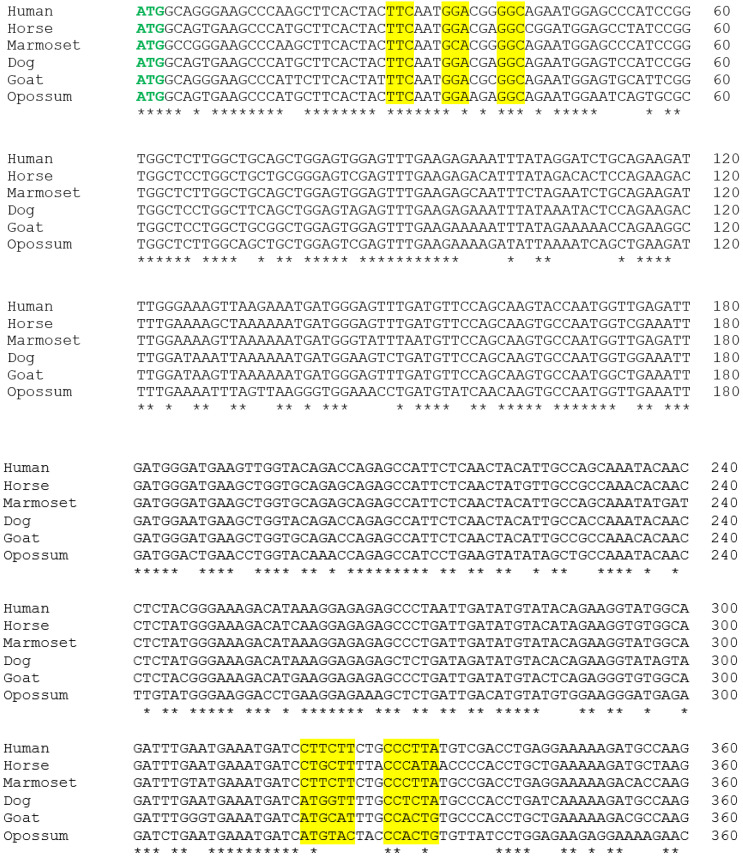
Aligned nucleotide sequences of the coding regions of *GSTA3* mRNAs from selected species including those from three reference species: Human (*Homo sapiens;* NM_000847.4), horse (*Equus caballus;* KC512384.1), and marmoset (*Callithrix jacchus*; XM_002746649), and those cloned from testes and sequenced in the present study: dog (*Canis lupus familiaris;* KJ651954), goat (*Capra hircus;* KM578828), and opossum (*Monodelphis domestica;* KM977823). Start and stop codons are identified in bold green and red type, respectively, and codons for the critical H-site (steroid binding) amino acids are highlighted. Asterisks under the alignment indicate conserved nucleotides across all six species.

**Figure 3 biomolecules-13-01420-f003:**
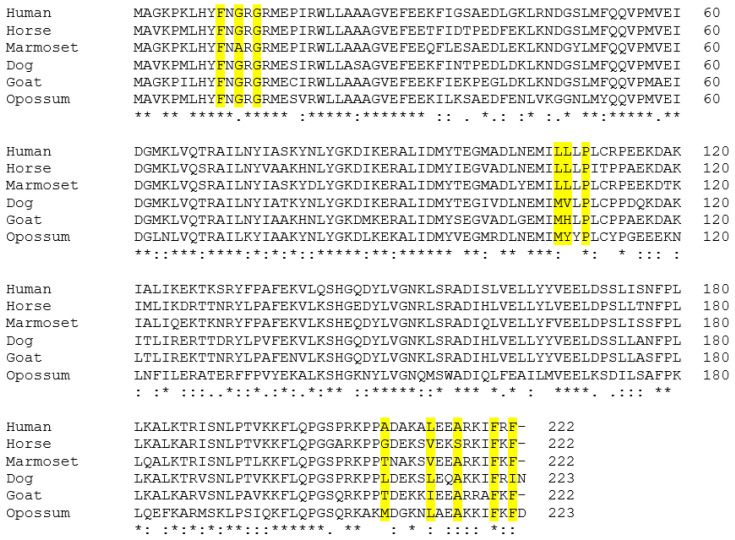
Aligned amino acid sequences for GSTA3 of all six species investigated including those used as references (translations of GenBank sequences in Figure 2). The critical amino acids for steroid isomerase activity of the human enzyme are highlighted. Symbols below the alignment: “*” identical residues, “:” similar residues, “.” conservation of residues with weakly similar properties in all six species GSTA3 proteins.

**Figure 4 biomolecules-13-01420-f004:**
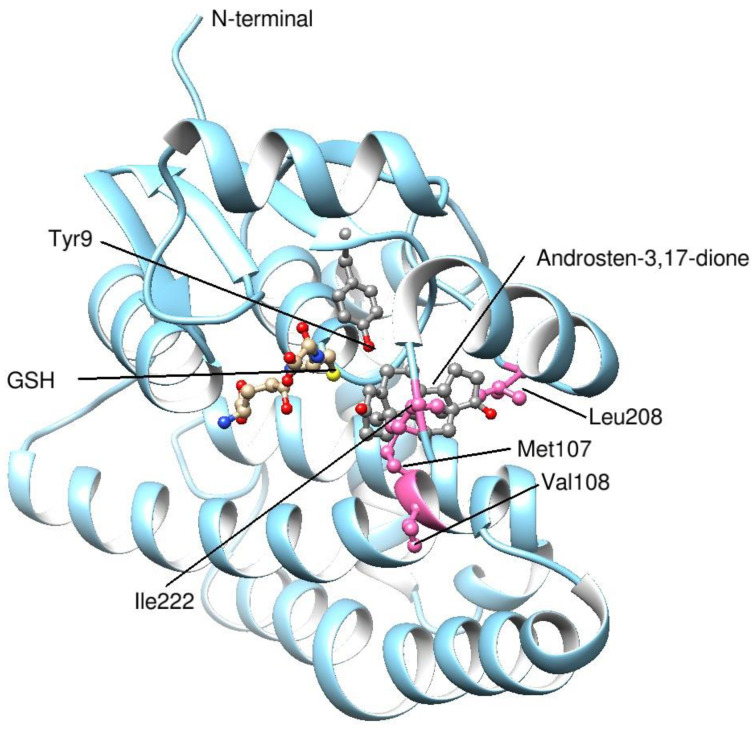
The dog GST A3-3 modeled with Δ^4^-androsten-3,17-dione bound in the active site. A homology structure of the dog enzyme was built in the program MODELLER (version 10.4, University of California San Franscisco, CA, USA) and depicted in Chimera 1.17.1 (version 1.17.1, University of California San Franscisco, CA, USA) based on the 85% amino acid sequence identity with the template human GST A3-3. The ligands glutathione and androsten-3,17-dione of the crystal structure of human GST A3-3 (PDB code 2vcv) are superpositioned in corresponding binding sites and the same binding modes in the model of the dog enzyme. In the model the sulfur of glutathione and the oxygen of Tyr9 implicated in the catalytic mechanism of steroid isomerization emerged adjacent to the scissile bonds of C4 and C6 in the androsten-3,17-dione. The four residues in the dog GST A3-3 differing from the corresponding H-site residues in the human enzyme are rendered in pink; the conserved H-site residues were overlapping in the two proteins.

**Figure 5 biomolecules-13-01420-f005:**
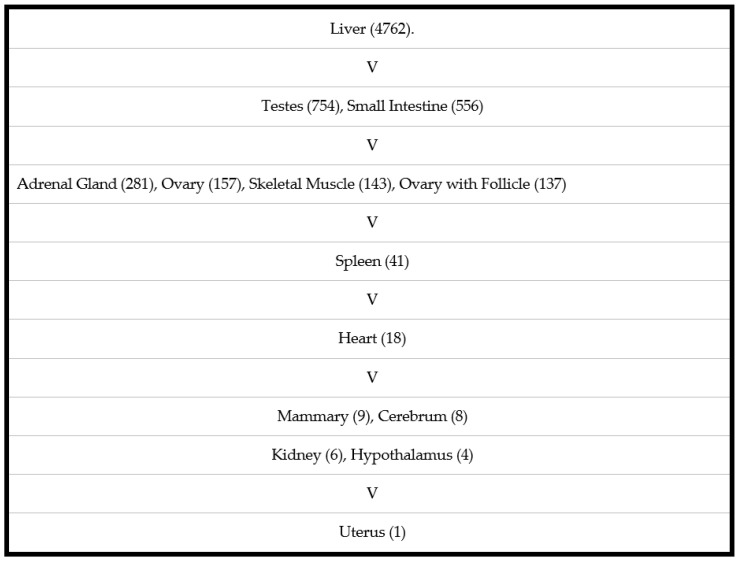
GSTA3 mRNA levels in canine tissues measured by real time quantitative polymerase chain reaction. The lowest value was set to 1.0 with all other values relative to that in order to rank the tissues based on their concentrations. “V” designates a 4-fold or more difference between tissue values above and below.

**Table 1 biomolecules-13-01420-t001:** PCR primer sequences (5′-3′) for TA cloning GSTA3 cDNAs into pCR2.1.

Species	Primer Set	Orientation	Sequence
Dog	Outside	Sense	GGAGACCTGCATCATGGCAGTGAAGCCCATG
		Antisense	AGGAGATTGGCCCTGCATGTGCT
	Inside	Sense	ATGGCGGGGAAGCCCAAGCTTCACTACTTCAATGG
		Antisense	CTGGGCATCCATTCCTGTTCAGTTAATCCT
Goat	Outside	Sense	GGAGACTGCATCATGGCAGTGAAGCCCATG
		Antisense	TCAAATTTGTCCCAAACAGCCCC
	Inside	Sense	ATGGCAGGGAAGCCCATTCTTCACTATTTCAATGG
		Antisense	CCCCGCCAGCCGCCAGCTTTATTAAAACTT
Opossum	Outside	Sense	GGAGACTGCCATCATGGCAGTGAAGCCCATG
		Antisense	TGTGTTTAAGAAACACAGAGTCA
	Inside	Sense	GGTAAAGAAGATATTCAAGGTTGATGA
		Antisense	TCATCAACCTTGAATATCTTCTTTACC

**Table 2 biomolecules-13-01420-t002:** PCR primer sequences (5′–3′) for cloning GSTA3 cDNAs into pET-21a+. Lowercase letters designate the restriction enzyme sites.

Species	Primer Set	Orientation	Sequence
Dog	Outside	Sense	CCAGAGACTACCATGGCGGGGAAGCCCAAG
		Antisense	TCTCAGGAGATTGGCCCTGCATG
	Inside	Sense	gcgaattcATGGCGGGGAAGCCCAAGCTTCACTACTTCAATGG
		Antisense	cgctcgagCTGGGCATCCATTCCTGTTCAGTTAATCCT
Goat	Outside	Sense	AGAACTGCTATTATGGCAGGGAAGCCCAT
		Antisense	TCAAATTTGTCCCAGACAGCCC
	Inside	Sense	gcgaattcATGGCAGGGAAGCCCATTCTTCACTATTTCAATGG
		Antisense	cgctcgagCCCCGCCAGCCGCCAGCTTTATTAAAACTT
Opossum	Outside	Sense	GAATGGAAGATCATGTCTGGGAAGCCCAT
		Antisense	TTGCATTACTTAGAACTCTTCCTGAATATTCAGCT
	Inside	Sense	gcgaattcGGTAAAGAAGATATTCAAGGTTGATGA
		Antisense	cgctcgagTCATCAACCTTGAATATCTTCTTTACC

**Table 3 biomolecules-13-01420-t003:** The percentages of identical nucleotides at each position between GSTA3 mRNAs of different mammalian species.

	Human	Marmoset	Dog	Horse	Goat	Opossum
Human	100	94	88	85	84	70
Marmoset	94	100	86	86	85	70
Dog	88	86	100	87	87	71
Horse	85	86	87	100	85	70
Goat	84	85	87	85	100	68
Opossum	70	70	71	70	68	100

**Table 4 biomolecules-13-01420-t004:** The percentages of identical amino acids at each position between GSTA3 proteins of different mammalian species.

	Human	Marmoset	Dog	Horse	Goat	Opossum
Human	100	88	85	81	81	64
Marmoset	88	100	79	82	79	63
Dog	85	79	100	81	83	62
Horse	81	82	81	100	82	61
Goat	81	79	83	82	100	62
Opossum	64	63	62	61	62	100

**Table 5 biomolecules-13-01420-t005:** Conservation of H-site residues involved in steroid isomerization across GSTA3 proteins of different mammalian species. The residues marked by “*” in column 1 were experimentally determined to be critical for the steroid isomerase activity of the human GSTA3 protein [9]. Identical amino acids at human GSTA3 residue positions are highlighted. Pig and cattle data are from [10,24] and GenBank accession numbers NP_999015.2 and NP_001071617.1, respectively.

Residue	Human	Marmoset	Horse	Dog	Goat	Opossum	Pig	Cattle
10 *	F	F	F	F	F	F	F	F
12 *	G	A	G	G	G	G	G	G
14	G	G	G	G	G	G	G	G
107	L	L	L	M	M	M	L	M
108	L	L	L	V	H	Y	L	H
110	P	P	P	P	P	P	P	P
111 *	L	L	L	L	L	L	L	L
208 *	A	T	G	L	T	M	T	T
213	L	V	V	L	I	L	L	I
216 *	A	A	S	A	A	V	A	A
220	F	F	F	F	F	F	F	F
222	F	F	F	I	F	V	F	F

## Data Availability

Experimental data are available from the authors upon request.

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
