# Peer review of "Conservation of Glutathione Transferase mRNA and Protein Sequences Similar to Human and Horse Alpha Class GST A3-3 across Dog, Goat, and Opossum Species"

_biomolecules, 2023, doi:10.3390/biom13091420_

Round 1
Reviewer 1 Report
Dear authors,
In the paper "Conservation of glutathione transferase mRNA and protein sequences similar to human and horse Alpha class GST A3-3 across dog, goat, and opossum species," the authors claim to have cloned these genes to explore the presence of GST A3-3 orthologs with high ketosteroid isomerase activity in various mammalian species.
Although this study has yielded three GST A3-3 homologs, it has yet to demonstrate that the gene is functional. Since it is only an uncertain estimate from the sequence conservation, the possibility that it is a pseudogene has not been ruled out. To serve the authors' purpose, the enzyme activity should be demonstrated in GST A3-3, or at least confirmed if steroid isomerase activity can be detected in the dog liver or testes where the gene expression is claimed to have been observed. Therefore, additional experiments should be revisited for this paper.
Page 10, Table 5
Although the enzyme activity is discussed based on the conservation of H-site residues, the discussion of activity based on this table needs to be done more carefully because the difference between the Pig enzyme, which is assumed to have deficient activity, and the human enzyme is a change in only 208, and Thr208 in the Pig enzyme is conserved in the Marmoset enzyme. The authors have yet to determine the enzyme activities. Nevertheless, the authors point out that Leu208 in the canine enzyme may interfere with the D-ring of the steroid substrate and inhibit enzyme activity. Is there a clear difference between Thr and Leu on the structural model?
Page 11, Line 21-26
The residue numbers of Leu108 and Met108 listed here may be 107.
Author Response
The authors thank the reviewers for their careful assessment of our manuscript entitled; “Conservation of glutathione transferase mRNA and protein sequences similar to human and horse Alpha class GST A3-3 across dog, goat, and opossum species.” The authors are grateful for the critical comments and suggestions that make this revision a much more clear and informational work. All of the reviewers’ comments (in italics below) have been responded to here and in the revised document.
Reviewer 1: In the paper "Conservation of glutathione transferase mRNA and protein sequences similar to human and horse Alpha class GST A3-3 across dog, goat, and opossum species," the authors claim to have cloned these genes to explore the presence of GST A3-3 orthologs with high ketosteroid isomerase activity in various mammalian species.
Although this study has yielded three GST A3-3 homologs, it has yet to demonstrate that the gene is functional. Since it is only an uncertain estimate from the sequence conservation, the possibility that it is a pseudogene has not been ruled out. To serve the authors' purpose, the enzyme activity should be demonstrated in GST A3-3, or at least confirmed if steroid isomerase activity can be detected in the dog liver or testes where the gene expression is claimed to have been observed. Therefore, additional experiments should be revisited for this paper.
The GSTA3 cDNAs were cloned (with reverse transcribed and PCR) from testes of dog, goat, and opossum species. The nucleotide and predicted amino acid sequences are very similar to those from other species (Figures 2 and 3). They are not pseudogenes for the following reasons: About 9% of human predicted pseudogenes are expressed, but their coding sequences accumulate mutations (frame shifts, premature stop codons) which prevent their translation (D.W. Thomsen and M. E. Dinger in Nature Reviews: Genetics 2016 doi: 10. 1038/nrg.2016.20). This was not the case for any of the three newly cloned GSTA3 mRNAs, which had open reading frames of the expected lengths and sequence similarity to the cognate clones from previously published species.
Page 10, Table 5
Although the enzyme activity is discussed based on the conservation of H-site residues, the discussion of activity based on this table needs to be done more carefully because the difference between the Pig enzyme, which is assumed to have deficient activity, and the human enzyme is a change in only 208, and Thr208 in the Pig enzyme is conserved in the Marmoset enzyme. The authors have yet to determine the enzyme activities. Nevertheless, the authors point out that Leu208 in the canine enzyme may interfere with the D-ring of the steroid substrate and inhibit enzyme activity. Is there a clear difference between Thr and Leu on the structural model?
We fully agree that expression of the cDNA sequences and characterization of the corresponding proteins is desirable. Unfortunately, the graduate students involved in the experimental work have finished their theses and are not available for further studies. Even more importantly, funding for the project is not currently available. The enzymological studies are therefore outside the scope of the present investigation and have to await new resources. Based on prior work on homologous cDNA sequences from several other biological species, we are still confident that the new GSTs represent bona fide enzymes. We believe that the new sequences are of value to the important area of GST research.
The pig GST A2-2 is actually a very active isomerase (27% of the equine activity). This was misrepresented has now been corrected in the text.
Page 11, Line 21-26
The residue numbers of Leu108 and Met108 listed here may be 107.
We appreciate that the mislabeling Leu108 and Met108 was detected on page 11; the labels have now been corrected to Leu107 and Met107.
With these changes, the authors think that the manuscript is now ready for publication.
Reviewer 2 Report
Authors presented new and high quality research data of high interest. The idea of work is clear. Way of solution is also clear (and rather traditional). Authors used different bioinformatics and experimental methods. Results look solid and reliable. All conclusions supported by experimental results. In general the manuscript can be published in present form except a few remrks.
1. Alignment of sequences (figures 2 and 3) should
2. I did not find information how 3D modeling was done. I recommend to use AlphaFold2 be done using font with fixed width
3. In figure 3 please show residues from specific sites. It will facilitate understanding for readers whish are new in this field.
4. I did not find description how 3D modeling was done. I recommend to use AlphaFold2 and it would be very nice to add table with RMSD between experimental and crystal structures of enzymes from different sources (like table 4 for primary structure homology)
Author Response
The authors thank the reviewers for their careful assessment of our manuscript entitled; “Conservation of glutathione transferase mRNA and protein sequences similar to human and horse Alpha class GST A3-3 across dog, goat, and opossum species.” The authors are grateful for the critical comments and suggestions that make this revision a much more clear and informational work. All of the reviewers’ comments (in italics below) have been responded to here and in the revised document.
Reviewer 2: Authors presented new and high quality research data of high interest. The idea of work is clear. Way of solution is also clear (and rather traditional). Authors used different bioinformatics and experimental methods. Results look solid and reliable. All conclusions supported by experimental results. In general the manuscript can be published in present form except a few remrks.
- Alignment of sequences (figures 2 and 3) should…These were redone to correct the formatting to a proportional font.
- I did not find information how 3D modeling was done. I recommend to use AlphaFold2 be done using font with fixed width…We think that the software we used for the modeling was adequate. We have added two references and a better description of the process and outcomes of it.
- In figure 3 please show residues from specific sites. It will facilitate understanding for readers whish are new in this field. Figure 3 was redone to show the residues.
- I did not find description how 3D modeling was done. I recommend to use AlphaFold2 and it would be very nice to add table with RMSD between experimental and crystal structures of enzymes from different sources (like table 4 for primary structure homology). We have added two references [28,29] and a better description of the process and outcomes of it. Please see 2.2.4 and scores generated by the modeling and the Figure 4 legend.
With these changes, the authors think that the manuscript is now ready for publication.
Round 2
Reviewer 1 Report
Dear authors,
The authors have tried their best to address every referee's comments. The referee recommends that this paper be published.